# Improving Inference for Neural Image Compression

**Yibo Yang, Robert Bamler, Stephan Mandt**
Department of Computer Science
University of California, Irvine
{yibo.yang, rbamler, mandt}@uci.edu

## Abstract

We consider the problem of lossy image compression with deep latent variable models. State-of-the-art methods [Ballé et al., 2018, Minnen et al., 2018, Lee et al., 2019] build on hierarchical variational autoencoders (VAEs) and learn inference networks to predict a compressible latent representation of each data point. Drawing on the variational inference perspective on compression [Alemi et al., 2018], we identify three approximation gaps which limit performance in the conventional approach: an amortization gap, a discretization gap, and a marginalization gap. We propose remedies for each of these three limitations based on ideas related to iterative inference, stochastic annealing for discrete optimization, and bits-back coding, resulting in the first application of bits-back coding to lossy compression. In our experiments, which include extensive baseline comparisons and ablation studies, we achieve new state-of-the-art performance on lossy image compression using an established VAE architecture, by changing only the inference method.

## 1 Introduction

Deep learning methods are reshaping the field of data compression, and recently started to outperform state-of-the-art classical codecs on image compression [Minnen et al., 2018]. Besides useful on its own, image compression is a stepping stone towards better video codecs [Lombardo et al., 2019, Habibian et al., 2019, Yang et al., 2020a], which can reduce a sizable amount of global internet traffic.

State-of-the-art neural methods for lossy image compression [Ballé et al., 2018, Minnen et al., 2018, Lee et al., 2019] learn a mapping between images and latent variables with a variational autoencoder (VAE). An *inference network* maps a given image to a compressible latent representation, which a generative network can then map back to a reconstructed image. In fact, compression can be more broadly seen as a form of inference: to compress—or "encode"—data, one has to perform inference over a well-specified decompression—or "decoding"—algorithm.

In classical compression codecs, the *de*coder has to follow a well-specified procedure to ensure interoperability between different implementations of the same codec. By contrast, the *en*coding process is typically not uniquely defined: different encoder implementations of the same codec often compress the same input data to different bitstrings. For example, `pngcrush` [Randers-Pehrson, 1997] typically produces smaller PNG files than `ImageMagick`. Yet, both programs are standard-compliant PNG encoder implementations as both produce a compressed file that decodes to the same image. In other words, both encoder implementations perform *correct* inference over the standardized decoder specification, but use different inference algorithms with different performance characteristics.

The insight that better inference leads to better compression performance even within the same codec motivates us to reconsider how inference is typically done in neural data compression with VAEs. In this paper, we show that the conventional *amortized* inference [Kingma and Welling, 2013, Rezende et al., 2014] in VAEs leaves substantial room for improvement when used for data compression.

We propose an improved inference method for data compression tasks with three main innovations:

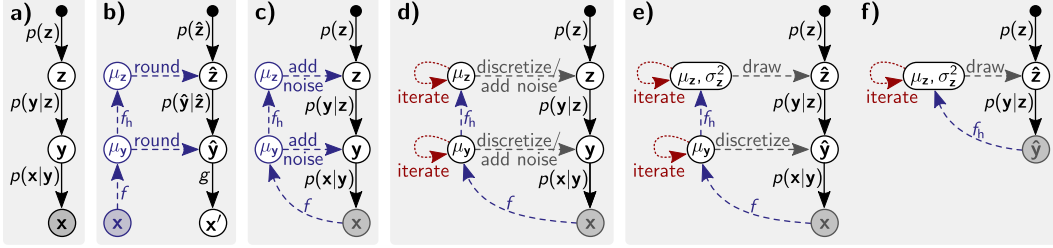

Figure 1: Graphical model and control flow charts. a) generative model with hyperlatents $\mathbf{z}$, latents $\mathbf{y}$, and image $\mathbf{x}$ [Minnen et al., 2018]; b) conventional method for compression (dashed blue, see Eq. 1) and decompression (solid black); c) common training objective due to [Ballé et al., 2017] (see Eq. 3); d) proposed hybrid amortized (dashed blue) / iterative (dotted red) inference (Section 3.1); e)-f) inference in the proposed lossy bitsback method (Section 3.3); the encoder first executes e) and then keeps $\hat{\mathbf{y}}$ fixed while executing f); the decoder reconstructs $\hat{\mathbf{y}}$ and then executes f) to get bits back.

1. *Improved amortization:* The amortized inference strategy in VAEs speeds up training, but is restrictive at compression time. We draw a connection between a recently proposed iterative procedure for compression [Campos et al., 2019] to the broader literature of VI that closes the amortization gap, which provides the basis of the following two novel inference methods.

2. *Improved discretization:* Compression requires discretizing the latent representation from VAEs, because only discrete values can be entropy coded. As inference over discrete variables is difficult, existing methods typically relax the discretization constraint in some way during inference and then discretize afterwards. We instead propose a novel method based on a stochastic annealing scheme that performs inference directly over discrete points.

3. *Improved entropy coding:* In lossless compression with latent-variable models, bits-back coding [Wallace, 1990, Hinton and Van Camp, 1993] allows approximately coding the latents with the *marginal* prior. It is so far believed that bits-back coding is incompatible with lossy compression [Habibian et al., 2019] because it requires inference on the *decoder* side, which does not have access to the exact (undistorted) input data. We propose a remedy to this limitation, resulting in the first application of bits-back coding to lossy compression.

We evaluate the above three innovations on an otherwise unchanged architecture of an established model and compare against a wide range of baselines and ablations. Our proposals significantly improve compression performance and results in a new state of the art in lossy image compression.

The rest of the paper is structured as follows: Section 2 summarizes lossy compression with VAEs, for which Section 3 proposes the above three improvements. Section 4 reports experimental results. We conclude in Section 5. We review related work in each relevant subsection.

## 2 Background: Lossy Neural Image Compression as Variational Inference

In this section, we summarize an existing framework for lossy image compression with deep latent variable models, which will be the basis of three proposed improvements in Section 3.

**Related Work.** Ballé et al. [2017] and Theis et al. [2017] were among the first to recognize a connection between the rate-distortion objective of lossy compression and the loss function of a certain kind of Variational Autoencoders (VAEs), and to apply it to end-to-end image compression. [Ballé et al., 2018] proposes a hierarchical model, upon which current state-of-the-art methods [Minnen et al., 2018, Lee et al., 2019] further improve by adding an auto-regressive component. For simplicity, we adopt the VAE of [Minnen et al., 2018] without the auto-regressive component, reviewed in this section below, which has also been a basis for recent compression research [Johnston et al., 2019].

**Generative Model.** Figure 1a shows the generative process of an image $\mathbf{x}$. The Gaussian likelihood $p(\mathbf{x}|\mathbf{y}) = \mathcal{N}(\mathbf{x}; g(\mathbf{y}; \theta), \sigma_{\mathbf{x}}^2 I)$ has a fixed variance $\sigma_{\mathbf{x}}^2$ and its mean is computed from latent variables $\mathbf{y}$ by a deconvolutional neural network (DNN) $g$ with weights $\theta$. A second DNN $g_{\mathrm{h}}$ with weights $\theta_{\mathrm{h}}$ outputs the parameters (location and scale) of the prior $p(\mathbf{y}|\mathbf{z})$ conditioned on "hyperlatents" $\mathbf{z}$.

**Compression and Decompression.**   Figure 1b illustrates compression ("encoding", dashed blue) and decompression ("decoding", solid black) as proposed in [Minnen et al., 2018]. The encoder passes a target image $\mathbf{x}$ through trained inference networks $f$ and $f_\mathrm{h}$ with weights $\phi$ and $\phi_\mathrm{h}$, respectively. The resulting continuous latent representations, $(\mu_\mathbf{y}, \mu_\mathbf{z})$, are rounded (denoted $\lfloor \cdot \rceil$) to discrete $(\hat{\mathbf{y}}, \hat{\mathbf{z}})$,

$$\hat{\mathbf{y}} = \lfloor \mu_\mathbf{y} \rceil; \quad \hat{\mathbf{z}} = \lfloor \mu_\mathbf{z} \rceil \qquad \text{where} \qquad \mu_\mathbf{y} = f(\mathbf{x}; \phi); \quad \mu_\mathbf{z} = f_\mathrm{h}(f(\mathbf{x}; \phi); \phi_\mathrm{h}). \tag{1}$$

The encoder then entropy-codes $\hat{\mathbf{z}}$ and $\hat{\mathbf{y}}$, using discretized versions $P(\hat{\mathbf{z}})$ and $P(\hat{\mathbf{y}}|\hat{\mathbf{z}})$ of the hyperprior $p(\mathbf{z})$ and (conditional) prior $p(\mathbf{y}|\mathbf{z})$, respectively, as entropy models. This step is simplified by a clever restriction on the shape of $p$, such that it agrees with the discretized $P$ on all integers (see [Ballé et al., 2017, 2018, Minnen et al., 2018]). The decoder then recovers $\hat{\mathbf{z}}$ and $\hat{\mathbf{y}}$ and obtains a lossy image reconstruction $\mathbf{x}' := \arg\max_\mathbf{x} p(\mathbf{x}|\hat{\mathbf{y}}) = g(\hat{\mathbf{y}}; \theta)$.

**Model Training.**   Let $\mathbf{x}$ be sampled from a training set of images. To train the above VAE for lossy compression, Theis et al. [2017], Minnen et al. [2018] consider minimizing a rate-distortion objective:

$$\mathcal{L}_\lambda(\lfloor \mu_\mathbf{y} \rceil, \lfloor \mu_\mathbf{z} \rceil) = \mathcal{R}(\lfloor \mu_\mathbf{y} \rceil, \lfloor \mu_\mathbf{z} \rceil) + \lambda \mathcal{D}(\lfloor \mu_\mathbf{y} \rceil, \mathbf{x})$$
$$= \underbrace{- \log_2 p(\lfloor \mu_\mathbf{z} \rceil)}_{\substack{\text{information content} \\ \text{of } \hat{\mathbf{z}} = \lfloor \mu_\mathbf{z} \rceil}} \underbrace{- \log_2 p(\lfloor \mu_\mathbf{y} \rceil \mid \lfloor \mu_\mathbf{z} \rceil)}_{\substack{\text{information content of} \\ \hat{\mathbf{y}} = \lfloor \mu_\mathbf{y} \rceil \text{ given } \hat{\mathbf{z}} = \lfloor \mu_\mathbf{z} \rceil}} + \lambda \underbrace{\left\lVert \mathbf{x} - g(\lfloor \mu_\mathbf{y} \rceil; \theta) \right\rVert_2^2}_{\text{distortion } \mathcal{D}} \tag{2}$$

where $(\mu_\mathbf{y}, \mu_\mathbf{z})$ is computed from the inference networks as in Eq. 1, and the parameter $\lambda > 0$ controls the trade-off between bitrate $\mathcal{R}$ under entropy coding, and distortion (reconstruction error) $\mathcal{D}$. As the rounding operations prevent gradient-based optimization, Ballé et al. [2017] propose to replace rounding during training by adding uniform noise from the interval $\left[-\frac{1}{2}, \frac{1}{2}\right]$ to each coordinate of $\mu_\mathbf{y}$ and $\mu_\mathbf{z}$ (see Figure 1c). This is equivalent to sampling from uniform distributions $q(\mathbf{y}|\mathbf{x})$ and $q(\mathbf{z}|\mathbf{x})$ with a fixed width of one centered around $\mu_\mathbf{y}$ and $\mu_\mathbf{z}$, respectively. One thus obtains the following relaxed rate-distortion objective, for a given data point $\mathbf{x}$,

$$\tilde{\mathcal{L}}_\lambda(\theta, \theta_\mathrm{h}, \phi, \phi_\mathrm{h}) = \mathbb{E}_{q(\mathbf{y}|\mathbf{x})\, q(\mathbf{z}|\mathbf{x})} \left[ - \log_2 p(\mathbf{z}) - \log_2 p(\mathbf{y}|\mathbf{z}) + \lambda \lVert \mathbf{x} - g(\mathbf{y}; \theta) \rVert_2^2 \right]. \tag{3}$$

**Connection to Variational Inference.**   As pointed out in [Ballé et al., 2017], the relaxed objective in Eq. 3 is the negative evidence lower bound (NELBO) of variational inference (VI) if we identify $\lambda = 1/(2\sigma_\mathbf{x}^2 \log 2)$. This draws a connection between lossy compression and VI [Blei et al., 2017, Zhang et al., 2019]. We emphasize a distinction between variational *inference* and variational *expectation maximization* (EM) [Beal and Ghahramani, 2003]: whereas variational EM trains a model (and employs VI as a subroutine), VI is used to compress data using a trained model. A central result of this paper is that improving inference in a fixed generative model at compression time already suffices to significantly improve compression performance.

## 3   Novel Inference Techniques for Data Compression

This section presents our main contributions. We identify three approximation gaps in VAE-based compression methods (see Section 2): an amortization gap, a discretization gap, and a marginalization gap. Recently, the amortization gap was considered by [Campos et al., 2019]; we expand on this idea in Section 3.1, bringing it under a wider framework of algorithms in the VI literature. We then propose two specific methods that improve inference at compression time: a novel inference method over discrete representations (Section 3.2) that closes the discretization gap, and a novel lossy bits-back coding method (Section 3.3) that closes the marginalization gap.

### 3.1   Amortization Gap and Hybrid Amortized-Iterative Inference

**Amortization Gap.**   Amortized variational inference [Kingma and Welling, 2013, Rezende et al., 2014] turns optimization over *local* (per-data) variational parameters into optimization over the *global* weights of an inference network. In the VAE of Section 2, the inference networks $f$ and $f_\mathrm{h}$ map an image $\mathbf{x}$ to local parameters $\mu_\mathbf{y}$ and $\mu_\mathbf{z}$ of the variational distributions $q(\mathbf{y}|\mathbf{x})$ and $q(\mathbf{z}|\mathbf{x})$, respectively (Eq. 1). Amortized VI speeds up training by avoiding an expensive inner inference loop of variational EM, but it leads to an amortization gap [Cremer et al., 2018, Krishnan et al., 2018], which is the difference between the value of the NELBO (Eq. 3) when $\mu_\mathbf{y}$ and $\mu_\mathbf{z}$ are obtained from the inference networks, compared to its true minimum when $\mu_\mathbf{y}$ and $\mu_\mathbf{z}$ are directly minimized. Since

the NELBO approximates the rate-distortion objective (Section 2), the amortization gap translates into sub-optimal performance when amortization is used at compression time. As discussed below, this gap can be closed by refining the output of the inference networks by iterative inference.

**Related Work.** The idea of combining amortized inference with iterative optimization has been studied and refined by various authors [Hjelm et al., 2016, Kim et al., 2018, Krishnan et al., 2018, Marino et al., 2018]. While not formulated in the language of variational autoencoders and variational inference, Campos et al. [2019] apply a simple version of this idea to compression. Drawing on the connection to hybrid amortized-iterative inference, we show that the method can be drastically improved by addressing the discretization gap (Section 3.2) and the marginalization gap (Section 3.3).

**Hybrid Amortized-Iterative Inference.** We reinterpret the proposal in [Campos et al., 2019] as a basic version of a hybrid amortized-iterative inference idea that changes inference only at compression time but not during model training. Figure 1d illustrates the approach. When compressing a target image $\mathbf{x}$, one initializes $\mu_\mathbf{y}$ and $\mu_\mathbf{z}$ from the trained inference networks $f$ and $f_\mathrm{h}$, see Eq. 1. One then treats $\mu_\mathbf{y}$ and $\mu_\mathbf{z}$ as *local* variational parameters, and minimizes the NELBO (Eq. 3) over $(\mu_\mathbf{y}, \mu_\mathbf{z})$ with the reparameterization trick and stochastic gradient descent. This approach thus separates inference *at test time* from inference *during model training*. We show in the next two sections that this simple idea forms a powerful basis for new inference approaches that drastically improve compression performance.

## 3.2 Discretization Gap and Stochastic Gumbel Annealing (SGA)

**Discretization Gap.** Compressing data to a bitstring is an inherently *discrete* optimization problem. As discrete optimization in high dimensions is difficult, neural compression methods instead optimize some relaxed objective function (such as the NELBO $\tilde{\mathcal{L}}_\lambda$ in Eq. 3) over continuous representations (such as $\mu_\mathbf{y}$ and $\mu_\mathbf{z}$), which are then discretized afterwards for entropy coding (see Eq. 1). This leads to a *discretization gap*: the difference between the true rate-distortion objective $\mathcal{L}_\lambda$ at the discretized representation $(\hat{\mathbf{y}}, \hat{\mathbf{z}})$, and the relaxed objective function $\tilde{\mathcal{L}}_\lambda$ at the continuous approximation $(\mu_\mathbf{y}, \mu_\mathbf{z})$.

**Related Work.** Current neural compression methods use a differentiable approximation to discretization *during model training*, such as Straight-Through Estimator (STE) [Bengio et al., 2013, Oord et al., 2017, Yin et al., 2019], adding uniform noise [Ballé et al., 2017] (see Eq. 3), stochastic binarization [Toderici et al., 2016], and soft-to-hard quantization [Agustsson et al., 2017]. Discretization *at compression time* was addressed in [Yang et al., 2020b] without assuming that the training procedure takes discretization into account, whereas our work makes this additional assumption.

**Stochastic Gumbel Annealing (SGA).** We refine the hybrid amortized-iterative inference approach of Section 3.1 to close the discretization gap: for a given image $\mathbf{x}$, we aim to find its *discrete* (in our case, integer) representation $(\hat{\mathbf{y}}, \hat{\mathbf{z}})$ that optimizes the rate-distortion objective $\mathcal{L}_\lambda$ in Eq. 2, this time re-interpreted as a function of $(\hat{\mathbf{y}}, \hat{\mathbf{z}})$ directly. Our proposed annealing scheme approaches this discretization optimization problem with the help of continuous proxy variables $\mu_\mathbf{y}$ and $\mu_\mathbf{z}$, which we initialize from the inference networks as in Eq. 1. We limit the following discussion to the latents $\mu_\mathbf{y}$ and $\hat{\mathbf{y}}$, treating the hyperlatents $\mu_\mathbf{z}$ and $\hat{\mathbf{z}}$ analogously. For each dimension $i$ of $\mu_\mathbf{y}$, we map the continuous coordinate $\mu_{\mathbf{y},i} \in \mathbb{R}$ to an integer coordinate $\hat{y}_i \in \mathbb{Z}$ by rounding either up or down. Let $r_{\mathbf{y},i}$ be a one-hot vector that indicates the rounding direction, with $r_{\mathbf{y},i} = (1,0)$ for rounding down (denoted $\lfloor \cdot \rfloor$) and $r_{\mathbf{y},i} = (0,1)$ for rounding up (denoted $\lceil \cdot \rceil$). Thus, the result of rounding is the inner product, $\hat{y}_i = r_{\mathbf{y},i} \cdot (\lfloor \mu_{\mathbf{y},i} \rfloor, \lceil \mu_{\mathbf{y},i} \rceil)$. Now we let $r_{\mathbf{y},i}$ be a Bernoulli variable with a "tempered" distribution

$$q_\tau(r_{\mathbf{y},i}|\mu_{\mathbf{y},i}) \propto \begin{cases} \exp\{-\psi(\mu_{\mathbf{y},i} - \lfloor \mu_{\mathbf{y},i} \rfloor)/\tau\} & \text{if } r_{\mathbf{y},i} = (1,0) \\ \exp\{-\psi(\lceil \mu_{\mathbf{y},i} \rceil - \mu_{\mathbf{y},i})/\tau\} & \text{if } r_{\mathbf{y},i} = (0,1) \end{cases} \quad (4)$$

with temperature parameter $\tau > 0$, and $\psi : [0,1) \to \mathbb{R}$ is an increasing function satisfying $\lim_{\alpha \to 1} \psi(\alpha) = \infty$ (this is chosen to ensure continuity of the resulting objective (5); we used $\psi = \tanh^{-1}$). Thus, as $\mu_{\mathbf{y},i}$ approaches an integer, the probability of rounding to that integer approaches one. Defining $q_\tau(r_\mathbf{y}|\mu_\mathbf{y}) = \prod_i q_\tau(r_{\mathbf{y},i}|\mu_{\mathbf{y},i})$, we thus minimize the stochastic objective

$$\tilde{\mathcal{L}}_{\lambda,\tau}(\mu_\mathbf{y}, \mu_\mathbf{z}) = \mathbb{E}_{q_\tau(r_\mathbf{y}|\mu_\mathbf{y})\,q_\tau(r_\mathbf{z}|\mu_\mathbf{z})}\Big[\mathcal{L}_\lambda\Big(r_\mathbf{y} \cdot (\lfloor \mu_\mathbf{y} \rfloor, \lceil \mu_\mathbf{y} \rceil), r_\mathbf{z} \cdot (\lfloor \mu_\mathbf{z} \rfloor, \lceil \mu_\mathbf{z} \rceil)\Big)\Big] \quad (5)$$

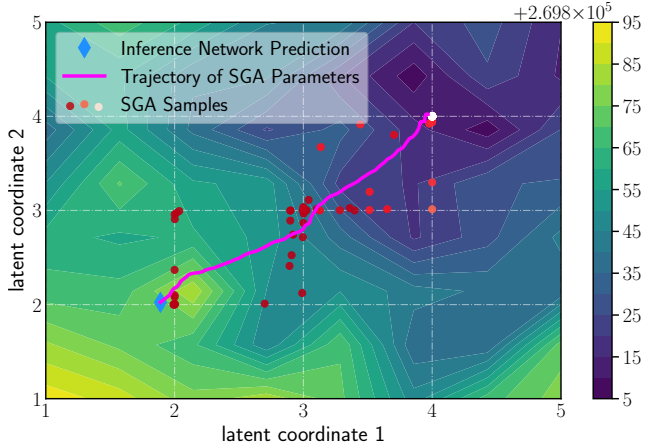
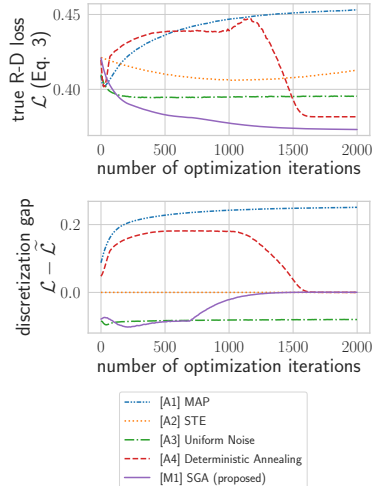

Figure 2: Visualizing SGA optimization. White dashed lines indicate the discretization grid. SGA samples are colored by temperature (lighter color corresponds to lower temperature). 10 Gumbel-softmax samples were used to approximate expectations in the objective (5), and temperature $\tau = 0.1$.

Figure 3: Comparing Stochastic Gumbel Annealing (Section 3.2) to alternatives (Table 1, [A1]-[A4]).

where we reintroduce the hyperlatents $\mathbf{z}$, and $r_{\mathbf{y}} \cdot (\lfloor \mu_{\mathbf{y}} \rfloor, \lceil \mu_{\mathbf{y}} \rceil)$ denotes the discrete vector $\hat{\mathbf{y}}$ obtained by rounding each coordinate $\mu_{\mathbf{y},i}$ of $\mu_{\mathbf{y}}$ according to its rounding direction $r_{\mathbf{y},i}$. At any temperature $\tau$, the objective in Eq. 5 smoothly interpolates the R-D objective in Eq. 2 between all integer points. We minimize Eq. 5 over $(\mu_{\mathbf{y}}, \mu_{\mathbf{z}})$ with stochastic gradient descent, propagating gradients through Bernoulli samples via the Gumbel-softmax trick [Jang et al., 2016, Maddison et al., 2016] using, for simplicity, the same temperature $\tau$ as in $q_{\tau}$. We note that in principle, REINFORCE [Williams, 1992] can also work instead of Gumbel-softmax. We anneal $\tau$ towards zero over the course of optimization, such that the Gumbel approximation becomes exact and the stochastic rounding operation converges to the deterministic one in Eq. 1. We thus name this method "Stochastic Gumbel Annealing" (SGA).

Figure 2 illustrates the loss function (Eq. 5) landscape of SGA near its converged solution, for a given Kodak image. Here, we visualize along the two coordinates of $\mu_{\mathbf{y}}$ whose optimization trajectories have the most variance among all coordinates, plotting the optimization trajectory of $\mu_{\mathbf{y}}$ (magenta line) and the stochastically rounded samples (red dots) from the inner product $r_{\mathbf{y}} \cdot (\lfloor \mu_{\mathbf{y}} \rfloor, \lceil \mu_{\mathbf{y}} \rceil)$. The final SGA sample converges to the deterministically rounded solution (white dot, $(4, 4)$), which is significantly better than the initialization predicted by inference network (blue diamond, near $(2, 2)$).

**Comparison to Alternatives.** Alternatively, we could have started with Eq. 2, and optimized it as a function of $(\mu_{\mathbf{y}}, \mu_{\mathbf{z}})$, using existing discretization methods to relax rounding; the optimized $(\mu_{\mathbf{y}}, \mu_{\mathbf{z}})$ would subsequently be rounded to $(\hat{\mathbf{y}}, \hat{\mathbf{z}})$. However, our method outperforms all these alternatives. Specifically, we tried (with shorthands [A1]-[A4] for 'ablation', referring to Table 1 of Section 4): [A1] **MAP**: completely ignoring rounding during optimization; [A2] **Straight-Through Estimator (STE)** [Bengio et al., 2013]: rounding only on the forward pass of automatic differentiation; [A3] **Uniform Noise** [Campos et al., 2019]: optimizing Eq. 3 over $(\mu_{\mathbf{y}}, \mu_{\mathbf{z}})$ at compression time, following the Hybrid Amortized-Iterative Inference approach of Section 3.1; and [A4] **Deterministic Annealing**: turning the stochastic rounding operations of SGA into deterministic weighted averages, by pushing the expectation operators w.r.t $q$ in Eq. 5 to *inside* the arguments of $\mathcal{L}_{\lambda}$, resulting in the loss function $\mathcal{L}_{\lambda}\left(\mathbb{E}_{q_{\tau}(r_{\mathbf{y}}|\mu_{\mathbf{y}})}[r_{\mathbf{y}} \cdot (\lfloor \mu_{\mathbf{y}} \rfloor, \lceil \mu_{\mathbf{y}} \rceil)], \mathbb{E}_{q_{\tau}(r_{\mathbf{z}}|\mu_{\mathbf{z}})}[r_{\mathbf{z}} \cdot (\lfloor \mu_{\mathbf{z}} \rfloor, \lceil \mu_{\mathbf{z}} \rceil)]\right)$ resembling [Agustsson et al., 2017].

We tested the above discretization methods on images from Kodak [Kodak], using a pre-trained hyper-prior model (Section 2). Figure 3 (left) shows learning curves for the true rate-distortion objective $\mathcal{L}_{\lambda}$, evaluated via rounding the intermediate $(\mu_{\mathbf{y}}, \mu_{\mathbf{z}})$ throughout optimization. Figure 3 (right) shows the discretization gap $\mathcal{L}_{\lambda} - \tilde{\mathcal{L}}_{\lambda}$, where $\tilde{\mathcal{L}}_{\lambda}$ is the relaxed objective of each method. Our proposed *SGA* method closes the discretization gap and achieves the lowest true rate-distortion loss among all methods compared, using the same initialization. *Deterministic Annealing* also closed the discretization gap, but converged to a worse solution. *MAP* naively converged to a non-integer solution, and *STE*

consistently diverged even with a tiny learning rate, as also observed by Yin et al. [2019] (following their results, we changed the gradient of the backward pass from identity to that of ReLU or clipped ReLU; neither helped). *Uniform Noise* produced a consistently negative discretization gap, as the noise-perturbed objective in Eq. 3 empirically overestimates the true rate-distortion objective $\mathcal{L}_\lambda$ on a large neighborhood around the initial $(\mu_{\mathbf{y}}, \mu_{\mathbf{z}})$ computed by the inference network; by contrast, SGA shrinks this gap by letting its variational distribution $q$ increasingly concentrate on integer solutions.

### 3.3   Marginalization Gap and Lossy Bits-Back Coding

**Marginalization Gap.**   To compress an image $\mathbf{x}$ with the hierarchical model of Section 2, ideally we would only need to encode and transmit its latent representation $\hat{\mathbf{y}}$ but not the hyper-latents $\hat{\mathbf{z}}$, as only $\hat{\mathbf{y}}$ is needed for reconstruction. This would require entropy coding with (a discretization of) the *marginal* prior $p(\mathbf{y}) = \int p(\mathbf{z}) \, p(\mathbf{y}|\mathbf{z}) \, d\mathbf{z}$, which unfortunately is computationally intractable. Therefore, the standard compression approach reviewed in Section 2 instead encodes some hyperlatent $\hat{\mathbf{z}}$ using the discretized hyperprior $P(\hat{\mathbf{z}})$ first, and then encodes $\hat{\mathbf{y}}$ using the entropy model $P(\hat{\mathbf{y}}|\hat{\mathbf{z}})$. This leads to a *marginalization gap*, which is the difference between the information content $-\log_2 P(\hat{\mathbf{z}}, \hat{\mathbf{y}}) = -\log_2 P(\hat{\mathbf{z}}) - \log_2 P(\hat{\mathbf{y}}|\hat{\mathbf{z}})$ of the transmitted tuple $(\hat{\mathbf{z}}, \hat{\mathbf{y}})$ and the information content of $\hat{\mathbf{y}}$ alone,

$$-\log_2 P(\hat{\mathbf{z}}, \hat{\mathbf{y}}) - (-\log_2 P(\hat{\mathbf{y}})) = -\log_2 P(\hat{\mathbf{z}}|\hat{\mathbf{y}}), \qquad (6)$$

i.e., the marginalization gap is the information content of the hyperlatent $\hat{\mathbf{z}}$ under the ideal posterior $P(\hat{\mathbf{z}}|\hat{\mathbf{y}})$, in the (discrete) latent generative process $\hat{\mathbf{z}} \to \hat{\mathbf{y}}$.

**Related Work.**   A similar marginalization gap has been addressed in *lossless* compression by bits-back coding [Wallace, 1990, Hinton and Van Camp, 1993] and made more practical by the BB-ANS algorithm [Townsend et al., 2019a]. Recent work has improved its efficiency in hierarchical latent variable models [Townsend et al., 2019b, Kingma et al., 2019], and extended it to flow models [Ho et al., 2019]. To our knowledge, bits-back coding has not yet been used in lossy compression. This is likely since bits-back coding requires Bayesian posterior inference on the *decoder* side, which seems incompatible with lossy compression, as the decoder does not have access to the undistorted data $\mathbf{x}$.

**Lossy Bits-Back Coding.**   We extend the bits-back idea to lossy compression by noting that the discretized latent representation $\hat{\mathbf{y}}$ is encoded losslessly (Figure 1b) and thus amenable to bits-back coding. A complication arises as bits-back coding requires that the encoder's inference over $\mathbf{z}$ can be exactly reproduced by the decoder (see below). This requirement is violated by the inference network in Eq. 1, where $\mu_{\mathbf{z}} = f_{\mathrm{h}}(f(\mathbf{x}; \phi); \phi_{\mathrm{h}})$ depends on $\mathbf{x}$, which is not available to the decoder in lossy compression. It turns out that the naive fix of setting instead $\mu_{\mathbf{z}} = f_{\mathrm{h}}(\hat{\mathbf{y}}; \phi_{\mathrm{h}})$ would hurt performance by more than what bits-back saves (see ablations in Section 4). We propose instead a two-stage inference algorithm that cuts the dependency on $\mathbf{x}$ after an initial joint inference over $\hat{\mathbf{y}}$ and $\mathbf{z}$.

Bits-back coding bridges the marginalization gap in Eq. 6 by encoding a limited number of bits of some additional side information (e.g., an image caption or a previously encoded image of a slide show) into the choice of $\hat{\mathbf{z}}$ using an entropy model $Q(\hat{\mathbf{z}}|\mu_{\mathbf{z}}, \sigma_{\mathbf{z}}^2)$ with parameters $\mu_{\mathbf{z}}$ and $\sigma_{\mathbf{z}}^2$. We obtain $Q(\hat{\mathbf{z}}|\mu_{\mathbf{z}}, \sigma_{\mathbf{z}}^2)$ by discretizing a Gaussian variational distribution $q(\mathbf{z}|\mu_{\mathbf{z}}, \sigma_{\mathbf{z}}^2) = \mathcal{N}(\mathbf{z}; \mu_{\mathbf{z}}, \mathrm{diag}(\sigma_{\mathbf{z}}^2))$, thus extending the last layer of the hyper-inference network $f_{\mathrm{h}}$ (Eq. 1) to output now a tuple $(\mu_{\mathbf{z}}, \sigma_{\mathbf{z}}^2)$ of means and diagonal variances. This replaces the form of variational distribution $q(\mathbf{z}|\mathbf{x})$ of Eq 3 as the restriction to a box-shaped distribution with fixed width is not necessary in bits-back coding. Additionally, we drop the restriction on the shape of the hyperprior $p(\mathbf{z})$ reviewed in Section 2, simply adopting the flexible density model as proposed in [Ballé et al., 2018] without convolving it with a uniform distribution, as $\mu_{\mathbf{z}}$ is no longer discretized to integers. We train the resulting VAE by minimizing the NELBO. Since $q(\mathbf{z}|\mu_{\mathbf{z}}, \sigma_{\mathbf{z}}^2)$ now has a variable width, the NELBO has an extra term compared to Eq. 3 that subtracts the entropy of $q(\mathbf{z}|\mu_{\mathbf{z}}, \sigma_{\mathbf{z}}^2)$, reflecting the expected number of bits we 'get back'. Thus, the NELBO is again a relaxed rate-distortion objective, where now the *net rate* is the compressed file size minus the amount of embedded side information.

Algorithm 1 describes lossy compression and decompression with the trained model. Subroutine encode initializes the variational parameters $\mu_{\mathbf{y}}$, $\mu_{\mathbf{z}}$, and $\sigma_{\mathbf{z}}^2$ conditioned on the target image $\mathbf{x}$ using the trained inference networks (line 1). It then jointly performs SGA over $\hat{\mathbf{y}}$ by following Section 3.2, and Black-Box Variational Inference (BBVI) over $\mu_{\mathbf{z}}$ and $\sigma_{\mathbf{z}}^2$ by minimizing the the NELBO (line 2). At the end of the routine, the encoder *decodes* the provided side information $\boldsymbol{\xi}$ (an arbitrary bitstring) into $\hat{\mathbf{z}}$ using $Q(\hat{\mathbf{z}}|\mu_{\mathbf{z}}, \sigma_{\mathbf{z}}^2)$ as entropy model (line 4) and then encodes $\hat{\mathbf{z}}$ and $\hat{\mathbf{y}}$ as usual (line 5).

**Algorithm 1:** Proposed lossy bits-back coding method (Section 3.3 and Figure 1e-f).

**Global Constants:** Trained hierarchical VAE with model $p(\mathbf{z}, \mathbf{y}, \mathbf{x}) = p(\mathbf{z})\, p(\mathbf{y}|\mathbf{z})\, p(\mathbf{x}|\mathbf{y})$ and
  inference networks $f(\,\cdot\,;\phi)$ and $f_\mathrm{h}(\,\cdot\,;\phi_\mathrm{h})$, see Figure 1e ($f$ is only used in subroutine `encode`).

**Subroutine** `encode`(image $\mathbf{x}$, side information $\boldsymbol{\xi}$) $\mapsto$ returns compressed bitstring $\mathbf{s}$

1 | Initialize $\mu_\mathbf{y} \leftarrow f(\mathbf{x};\phi)$ and $(\mu_\mathbf{z}, \sigma_\mathbf{z}^2) \leftarrow f_\mathrm{h}(\mu_\mathbf{y};\phi_\mathrm{h})$. &emsp;&emsp;&emsp; ▷ *Figure 1e (blue)*
2 | Optimize over $\hat{\mathbf{y}}$ using SGA (Section 3.2), and over $\mu_\mathbf{z}$ and $\sigma_\mathbf{z}^2$ using BBVI. &emsp; ▷ *Figure 1e (red)*
3 | Reset $(\mu_\mathbf{z}, \sigma_\mathbf{z}^2) \leftarrow$ `reproducible_BBVI`($\hat{\mathbf{y}}$). &emsp;&emsp;&emsp;&emsp;&emsp;&emsp; ▷ *See below.*
4 | Decode side information $\boldsymbol{\xi}$ into $\hat{\mathbf{z}}$ using $Q(\hat{\mathbf{z}}|\mu_\mathbf{z}, \sigma_\mathbf{z}^2)$ as entropy model.
5 | Encode $\hat{\mathbf{z}}$ and $\hat{\mathbf{y}}$ into $\mathbf{s}$ using $P(\hat{\mathbf{z}})$ and $P(\hat{\mathbf{y}}|\hat{\mathbf{z}})$ as entropy models, respectively.

**Subroutine** `decode`(compressed bitstring $\mathbf{s}$) $\mapsto$ returns (lossy reconstruction $\mathbf{x}'$, side info $\boldsymbol{\xi}$)

6 | Decode $\mathbf{s}$ into $\hat{\mathbf{z}}$ and $\hat{\mathbf{y}}$; then get reconstructed image $\mathbf{x}' = \arg\max_\mathbf{x} p(\mathbf{x}|\hat{\mathbf{y}})$.
7 | Set $(\mu_\mathbf{z}, \sigma_\mathbf{z}^2) \leftarrow$ `reproducible_BBVI`($\hat{\mathbf{y}}$). &emsp;&emsp;&emsp;&emsp;&emsp;&emsp; ▷ *See below.*
8 | Encode $\hat{\mathbf{z}}$ into $\boldsymbol{\xi}$ using $Q(\hat{\mathbf{z}}|\mu_\mathbf{z}, \sigma_\mathbf{z}^2)$ as entropy model.

**Subroutine** `reproducible_BBVI`(discrete latents $\hat{\mathbf{y}}$) $\mapsto$ returns variational parameters $(\mu_\mathbf{z}, \sigma_\mathbf{z}^2)$

9 | Initialize $(\mu_\mathbf{z}, \sigma_\mathbf{z}^2) \leftarrow f_\mathrm{h}(\hat{\mathbf{y}};\phi_\mathrm{h})$; seed random number generator reproducibly. ▷ *Figure 1f (blue)*
10 | Refine $\mu_\mathbf{z}$ and $\sigma_\mathbf{z}^2$ by running BBVI for fixed $\hat{\mathbf{y}}$. &emsp;&emsp;&emsp;&emsp; ▷ *Figure 1f (red)*

The important step happens on line 3: up until this step, the fitted variational parameters $\mu_\mathbf{z}$ and $\sigma_\mathbf{z}^2$ depend on the target image $\mathbf{x}$ due to their initialization on line 1. This would prevent the decoder, which does not have access to $\mathbf{x}$, from reconstructing the side information $\boldsymbol{\xi}$ by encoding $\hat{\mathbf{z}}$ with the entropy model $Q(\hat{\mathbf{z}}|\mu_\mathbf{z}, \sigma_\mathbf{z}^2)$. Line 3 therefore re-fits the variational parameters $\mu_\mathbf{z}$ and $\sigma_\mathbf{z}^2$ in a way that is exactly reproducible based on $\hat{\mathbf{y}}$ alone. This is done in the subroutine `reproducible_BBVI`, which performs BBVI in the prior model $p(\mathbf{z}, \mathbf{y}) = p(\mathbf{z})p(\mathbf{y}|\mathbf{z})$, treating $\mathbf{y} = \hat{\mathbf{y}}$ as observed and only $\mathbf{z}$ as latent. Although we reset $\mu_\mathbf{z}$ and $\sigma_\mathbf{z}^2$ on line 3 immediately after optimizing over them on line 2, optimizing jointly over both $\hat{\mathbf{y}}$ and $(\mu_\mathbf{z}, \sigma_\mathbf{z}^2)$ on line 2 allows the method to find a better $\hat{\mathbf{y}}$.

The decoder decodes $\hat{\mathbf{z}}$ and $\hat{\mathbf{y}}$ as usual (line 6). Since the subroutine `reproducible_BBVI` depends only on $\hat{\mathbf{y}}$ and uses a fixed random seed (line 9), calling it from the decoder on line 7 yields the exact same entropy model $Q(\hat{\mathbf{z}}|\mu_\mathbf{z}, \sigma_\mathbf{z}^2)$ as used by the encoder, allowing the decoder to recover $\boldsymbol{\xi}$ (line 8).

## 4   Experiments

We demonstrate the empirical effectiveness of our approach by applying two variants of it —with and without bits-back coding—to an established (but not state-of-the-art) base model [Minnen et al., 2018]. We improve its performance drastically, achieving an average of over $15\%$ BD rate savings on Kodak and $20\%$ on Tecnick [Asuni and Giachetti, 2014], outperforming the previous state-of-the-art of both classical and neural lossy image compression methods. We conclude with ablation studies.

**Proposed Methods.**   Table 1 describes all compared methods, marked as [M1]-[M9] for short. We tested two variants of our method ([M1] and [M2]): *SGA* builds on the exact same model and training procedure as in [Minnen et al., 2018] (the *Base Hyperprior* model [M3]) and changes only how the trained model is used for compression by introducing hybrid amortized-iterative inference [Campos et al., 2019] (Section 3.1) and Stochastic Gumbel Annealing (Section 3.2). *SGA+BB* adds to it bits-back coding (Section 3.3), which requires changing the inference model over hyperlatents $\mathbf{z}$ to admit a more flexible variational distribution $q(\mathbf{z}|\mathbf{x})$, and no longer restricts the shape of $p(\mathbf{z})$, as discussed in Section 3.3. The two proposed variants address different use cases: *SGA* is a "standalone" variant of our method that compresses individual images, while *SGA+BB* encodes images with additional side information. In all results, we used Adam [Kingma and Ba, 2015] for optimization, and annealed the temperature of SGA by an exponential decay schedule, and found good convergence without per-model hyperparameter tuning. We provide details in the Supplementary Material.

**Baselines.**   We compare to the *Base Hyperprior* model from [Minnen et al., 2018] without our proposed improvements ([M3] in Table 1), two state-of-the-art neural methods ([M4] and [M5]), two

Table 1: Compared methods ([M1]-[M9]) and ablations ([A1]-[A6]). We propose two variants: [M1] compresses images, and [M2] compresses images + side information via bits-back coding.

| | Name | Explanation and Reference (for baselines) |
|---|---|---|
| **ours** | [M1] SGA | Proposed standalone variant: same trained model as [M3], SGA (Section 3.2) at compression time. |
| | [M2] SGA + BB | Proposed variant with bits-back coding: both proposed improvements of Sections 3.2, and 3.3. |
| **baselines** | [M3] Base Hyperprior | Base method of our two proposals, reviewed in Section 2 of the present paper [Minnen et al., 2018]. |
| | [M4] Context + Hyperprior | Like [M3] but with an extra context model defining the prior $p(\mathbf{y}|\mathbf{z})$ [Minnen et al., 2018]. |
| | [M5] Context-Adaptive | Context-Adaptive Entropy Model proposed in [Lee et al., 2019]. |
| | [M6] Hyperprior Scale-Only | Like [M3] but the hyperlatents $\mathbf{z}$ model only the scale (not the mean) of $p(\mathbf{y}|\mathbf{z})$ [Ballé et al., 2018]. |
| | [M7] CAE | Pioneering "Compressive Autoencoder" model proposed in [Theis et al., 2017]. |
| | [M8] BPG 4:4:4 | State-of-the-art classical lossy image compression codec 'Better Portable Graphics' [Bellard, 2014]. |
| | [M9] JPEG 2000 | Classical lossy image compression method [Adams, 2001]. |
| **ablations** | [A1] MAP | Like [M1] but with continuous optimization over $\mu_{\mathbf{y}}$ and $\mu_{\mathbf{z}}$ followed by rounding instead of SGA. |
| | [A2] STE | Like [M1] but with straight-through estimation (i.e., round only on backpropagation) instead of SGA. |
| | [A3] Uniform Noise | Like [M1] but with uniform noise injection instead of SGA (see Section 3.1) [Campos et al., 2019]. |
| | [A4] Deterministic Annealing | Like [M1] but with deterministic version of Stochastic Gumbel Annealing |
| | [A5] BB without SGA | Like [M2] but without optimization over $\hat{\mathbf{y}}$ at compression time. |
| | [A6] BB without iterative inference | Like [M2] but without optimization over $\hat{\mathbf{y}}$, $\mu_{\mathbf{z}}$, and $\sigma_{\mathbf{z}}^2$ at compression time. |

other neural methods ([M6] and [M7]), the state-of-the art classical codec BPG [M8], and JPEG 2000 [M9]. We reproduced the *Base Hyperprior* results, and took the other results from [Ballé et al.].

**Results.** Figure 4 compares the compression performance of our proposed method to existing baselines on the Kodak dataset, using the standard Peak Signal-to-Noise Ratio (PSNR) quality metric (higher is better), averaged over all images for each considered quality setting $\lambda \in [0.001, 0.08]$. The bitrates in our own results ([M1-3], [A1-6]) are based on rate estimates from entropy models, but do not differ significantly from file sizes produced by our codec implementation (which has negligible ($< 0.5\%$) overhead). The left panel of Figure 4 plots PSNR vs. bitrate, and the right panel shows the resulting BD rate savings [Bjontegaard, 2001] computed relative to BPG as a function of PSNR for readability (higher is better). The BD plot cuts out CAE [M7] and JPEG 2000 [M9] at the bottom, as they performed much worse. Overall, both variants of the proposed method (blue and orange lines in Figure 4) improve substantially over the *Base Hyperprior* model (brown), and outperform the previous state-of-the-art. We report similar results on the Tecnick dataset [Asuni and Giachetti, 2014] in the Supplementary Material. Figure 6 shows a qualitative comparison of a compressed image (in the order of BPG, *SGA* (proposed), and the *Base Hyperprior*), in which we see that our method notably enhances the baseline image reconstruction at comparable bitrate, while avoiding unpleasant visual artifacts of BPG.

**Ablations.** Figure 5 compares BD rate improvements of the two variants of our proposal (blue and orange) to six alternative choices (ablations [A1]-[A6] in Table 1), measured relative to the *Base Hyperprior* model [Minnen et al., 2018] (zero line; [M3] in Table 1), on which our proposals build. [A1]-[A4] replace the discretization method of *SGA* [M1] with four alternatives discussed at the end of Section 3.2. [A5] and [A6] are ablations of our proposed bits-back variant *SGA+BB* [M2], which remove iterative optimization over $\hat{\mathbf{y}}$, or over $\hat{\mathbf{y}}$ and $q(\mathbf{z})$, respectively, at compression time. Going from red [A3] to orange [A2] to blue [M1] traces our proposed improvements over [Campos et al., 2019] by adding SGA (Section 3.2) and then bits-back (Section 3.3). It shows that iterative inference

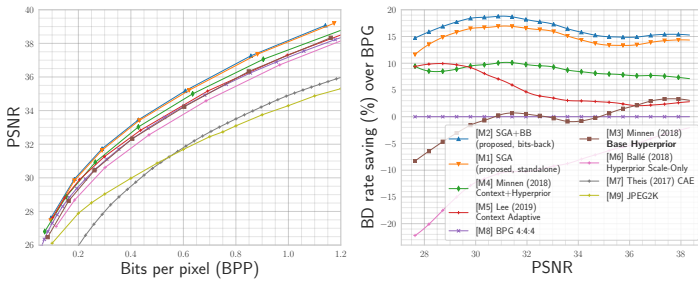
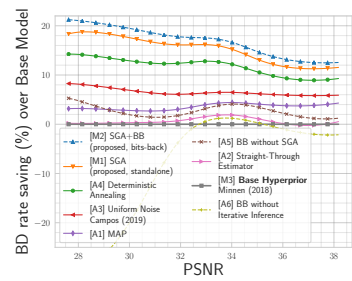

Figure 4: Compression performance comparisons on Kodak against existing baselines. Left: R-D curves. Right: BD rate savings (%) relative to BPG. Legend shared; higher values are better in both.

Figure 5: BD rate savings of various ablation methods relative to *Base Hyperprior* on Kodak.

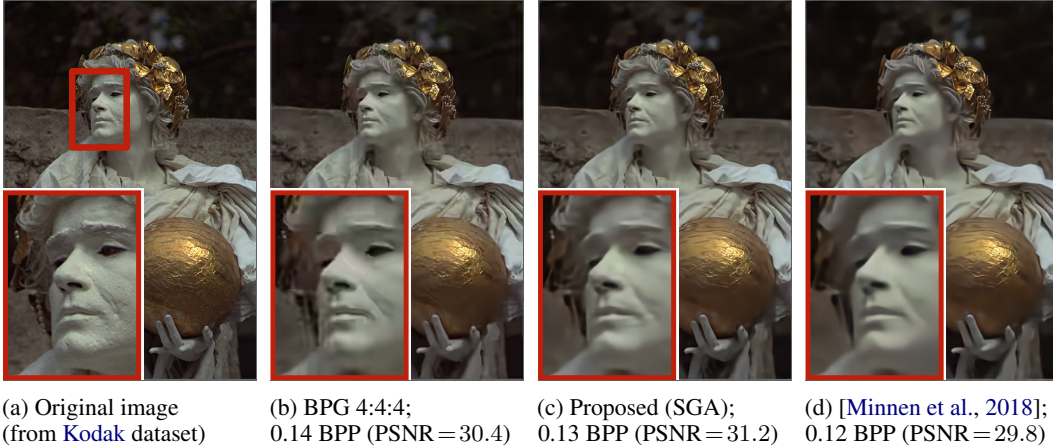

(a) Original image
(from Kodak dataset)

(b) BPG 4:4:4;
0.14 BPP (PSNR = 30.4)

(c) Proposed (SGA);
0.13 BPP (PSNR = 31.2)

(d) [Minnen et al., 2018];
0.12 BPP (PSNR = 29.8)

Figure 6: Qualitative comparison of lossy compression performance. Our method (c; [M1] in Table 1) significantly boosts the visual quality of the *Base Hyperprior* method (d; [M3] in Table 1) at similar bit rates. It sharpens details of the hair and face (see insets) obscured by the baseline method (d), while avoiding the ringing artifacts around the jaw and ear pendant produced by a classical codec (b).

(Section 3.1) and SGA contribute approximately equally to the performance gain, whereas the gain from bits-back coding is smaller. Interestingly, lossy bits-back coding without iterative inference and SGA actually hurts performance [A6]. As Section 3.3 mentioned, this is likely because bits-back coding constrains the posterior over $\mathbf{z}$ to be conditioned only on $\hat{\mathbf{y}}$ and not on the original image $\mathbf{x}$.

# 5  Discussion

Starting from the variational inference view on data compression, we proposed three enhancements to the standard inference procedure in VAEs: hybrid amortized-iterative inference, Stochastic Gumbel Annealing, and lossy bits-back coding, which translated to dramatic performance gains on lossy image compression. Improved inference provides a new promising direction to improved compression, orthogonal to modeling choices (e.g., with auto-regressive priors [Minnen et al., 2018], which can harm decoding efficiency). Although lossy-bits-back coding in the present VAE only gave relatively minor benefits, it may reach its full potential in more hierarchical architectures as in [Kingma et al., 2019]. Similarly, carrying out iterative inference also at *training time* may lead to even more performance improvement, with techniques like iterative amortized inference [Marino et al., 2018].

## Broader Impacts

Improved image and video compression is becoming more and more important as our lives become more digital. For instance, video compression is of enormous societal relevance, as over 80% of web traffic is due to video streaming, and the share of video data is expected to increase even further in the future [Cisco, 2017]. Moreover, there is an explosion of different new data formats that need to be efficiently compressed; examples include high resolution medical images, LIDAR data, 3D graphics data, DNA sequences, etc. It would be very costly to design custom codecs for such data, making learnable compression indispensable. Ultimately, better compression algorithms, both for data and models, will also facilitate the possibility to carry-out machine learning on local devices, as opposed to large centralized servers. This may address many of our privacy concerns.

Currently, neural compression comes with a few drawbacks compared to hand-engineered codecs, such as higher computation/storage/energy cost, and lack of robustness, which we hope can be overcome by active research in this area. Our work on improving the encoding procedure already has the potential to lower the *overall* resource consumption of a neural codec, for applications where the increased cost of encoding can be amortized across many file transfer/decoding operations and translate to net savings, e.g., on a content hosting website that has millions of visitors per day.

## Acknowledgments and Disclosure of Funding

We thank Yang Yang for valuable feedback on the manuscript. Yibo Yang acknowledges funding from the Hasso Plattner Foundation. This material is based upon work supported by the Defense Advanced Research Projects Agency (DARPA) under Contract No. HR001120C0021. Any opinions, findings and conclusions or recommendations expressed in this material are those of the author(s) and do not necessarily reflect the views of the Defense Advanced Research Projects Agency (DARPA). Furthermore, this work was supported by the National Science Foundation under Grants 1928718, 2003237 and 2007719, and by Qualcomm. Stephan Mandt consulted for Disney and Google.

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
