[Supplementary Material]

# Supplementary Material to
# Improving Inference for Neural Compression

## S1 Stochastic Annealing

Here we provide conceptual illustrations of our stochastic annealing idea on a simple example.

Consider the following scalar optimization problem over integers,

$$\underset{z \in \mathbb{Z}}{\text{minimize}} \quad f(z) = z^2$$

Following our stochastic annealing method in Section. 3.2, we let $x \in \mathbb{R}$ be a continuous proxy variable, and $r \in \{(1,0), (0,1)\}$ be the one-hot vector of stochastic rounding direction. We let $r$ follow a Bernoulli distribution with temperature $\tau > 0$,

$$q_\tau(r|x) = \begin{cases} \exp\{-\psi(x - \lfloor x \rfloor)/\tau\}/C(x,\tau) & \text{if } r = (1,0) \\ \exp\{-\psi(\lceil x \rceil - x)/\tau\}/C(x,\tau) & \text{if } r = (0,1) \end{cases}$$

where

$$C(x,\tau) = \exp\{-\psi(x - \lfloor x \rfloor)/\tau\} + \exp\{-\psi(\lceil x \rceil - x)/\tau\}$$

is the normalizing constant.

The stochastic optimization problem for SGA is then

$$\underset{x \in \mathbb{R}}{\text{minimize}} \quad \ell_\tau(x) := \mathbb{E}_{q_\tau(r|x)}[f(r \, \cdot \, (\lfloor x \rfloor, \lceil x \rceil)]$$

$$= \frac{\exp\{-\psi(x - \lfloor x \rfloor)/\tau\}}{C(x,\tau)} f(\lfloor x \rfloor) + \frac{\exp\{-\psi(\lceil x \rceil - x)/\tau\}}{C(x,\tau)} f(\lceil x \rceil)$$

Below we plot the rounding probability $q_\tau(r = (0,1)|x)$ and the SGA objective $\ell_\tau$ as a function of $x$, at various temperatures.

Figure S1: Illustration of the probability $q_\tau(r = (0,1)|x)$ of rounding *up*, on the interval $(0,1)$, at various temperatures.

Figure S2: Graphs of stochastic annealing objective $\ell_\tau$ at various temperatures. $\ell_\tau$ interpolates the discrete optimization problem between integers, and approaches $\text{round}(x)^2$ as $\tau \to 0$.

## S2 Model Architecture and Training

As mentioned in the main text, the *Base Hyperprior* model uses the same architecture as in Table 1 of Minnen et al. [2018] except without the "Context Prediction" and "Entropy Parameters" components (this model was referred to as "Mean & Scale Hyperprior" in this work). The model is mostly already implemented in `bmshj2018.py`[1] from Ballé et al., Ballé et al. [2018], and we modified their implementation to double the number of output channels of the `HyperSynthesisTransform` to predict both the mean and (log) scale of the conditional prior $p(\mathbf{y}|\mathbf{z})$ over the latents $\mathbf{y}$.

As mentioned in Section 3.3 and 4, lossy bits-back modifies the above *Base Hyperprior* as follows:

1. The number of output channels of the hyper inference network is doubled to compute both the mean and diagonal (log) variance of Gaussian $q(\mathbf{z}|\mathbf{x})$;

2. The hyperprior $p(\mathbf{z})$ is no longer restricted to the form of a flexible density model convolved with the uniform distribution on $[-0.5, 0.5]$; instead it simply uses the flexible density, as described in the Appendix of Ballé et al. [2018].

We trained our models (both the *Base Hyperprior* and the bits-back variant) on CLIC-2018 [2] images using minibatches of eight $256 \times 256$ randomly-cropped image patches, for $\lambda \in \{0.001, 0.0025, 0.005, 0.01, 0.02, 0.04, 0.08\}$. Following Ballé et al. [2018], we found that increasing the number of latent channels helps avoid the "bottleneck" effect and improve rate-distortion performance at higher rates, and increased the number of latent channels from 192 to 256 for models trained with $\lambda = 0.04$ and 0.08. All models were trained for 2 million steps, except the ones with $\lambda = 0.001$ which were trained for 1 million steps, and $\lambda = 0.08$ which were trained for 3 million steps. Our code and pre-trained models can be found at `https://github.com/mandt-lab/improving-inference-for-neural-image-compression`.

## S3   Hyper-Parameters of Various Methods Considered

Given a pre-trained model (see above) and image(s) $\mathbf{x}$ to be compressed, our experiments in Section 4 explored hybrid amortized-iterative inference to improve compression performance. Below we provide hyper-parameters of each method considered:

**Stochastic Gumbel Annealing.**   In all experiments using SGA (including the *SGA+BB* experiments, which optimized over $\hat{\mathbf{y}}$ using SGA), we used the Adam optimizer with an initial learning rate of 0.005, and an exponentially decaying temperature schedule $\tau(t) = \min(\exp\{-ct\}, 0.5)$, where $t$ is the iteration number, and $c > 0$ controls the speed of the decay. We found that lower $c$ generally increases the number of iterations needed for convergence, but can also yield slightly better solutions; in all of our experiments we set $c = 0.001$, and obtained good convergence with 2000 iterations.

**Temperature Annealing Schedule for SGA.**   We initially used a naive temperature schedule, $\tau(t) = \tau_0 \exp\{-ct\}$ for SGA, where $\tau_0$ is the initial temperature (typically set to 0.5 to simulate soft quantization), and found that aggressive annealing with a large decay factor $c$ can lead to suboptimal solutions, as seen with $c = 0.002$ or $c = 0.001$ in the left subplot of Figure S3. We found that we can overcome the suboptimality from fast annealing by an initial stage of optimization at a fixed temperature before annealing: in the initial stage, we fix the temperature to some relatively high value $\tau_0$ (to simulate soft discretization), and run the optimization for $t_0$ steps such that the R-D objective roughly converges. We demonstrate this in the right subplot of Figure S3, where we modify the naive schedule to $\tau(t) = \min\{\tau_0, \tau_0 \exp\{-c(t - t_0)\}\}$, so that SGA roughly converges after $t_0 = 700$ steps at temperature $\tau_0 = 0.5$ before annealing. As can be seen, the results of SGA are robust to different choices of decay factor $c$, with $c = 0.002$ and $c = 0.001$ giving comparably good results to $c = 0.0005$ but with faster convergence.

Figure S3: Comparing different annealing schedules for SGA (Section 3.2).

**Lossy bits-back.**   In the *SGA+BB* experiments, line 2 of the `encode` subroutine of Algorithm 1 performs joint optimization w.r.t. $\mu_{\mathbf{y}}$, $\mu_{\mathbf{z}}$, and $\sigma_{\mathbf{z}}^2$ with Black-box Variational Inference (BBVI), using

the reparameterization trick to differentiate through Gumbel-softmax samples for $\mu_{\mathbf{y}}$, and Gaussian samples for $(\mu_{\mathbf{z}}, \sigma_{\mathbf{z}}^2)$. We used Adam with an initial learning rate of 0.005, and ran 2000 stochastic gradient descent iterations; the optimization w.r.t. $\mu_{\mathbf{y}}$ (SGA) used the same temperature annealing schedule as above. The `reproducible_BBVI` subroutine of Algorithm 1 used Adam with an initial learning rate of 0.003 and 2000 stochastic gradient descent iterations; we used the same random seed for the encoder and decoder for simplicity, although a hash of $\hat{\mathbf{y}}$ can also be used instead.

**Alternative Discretization Methods**    In Figure 3 and ablation studies of Section 4, all alternative discretization methods were optimized with Adam for 2000 iterations to compare with *SGA*. The initial learning rate was 0.005 for *MAP*, *Uniform Noise*, and *Deterministic Annealing*, and 0.0001 for *STE*. All methods were tuned on a best-effort basis to ensure convergence, except that *STE* consistently encountered convergence issues even with a tiny learning rate (see [Yin et al., 2019]). The rate-distortion results for *MAP* and *STE* were calculated with early stopping (i.e., using the intermediate $(\hat{\mathbf{y}}, \hat{\mathbf{z}})$ with the lowest true rate-distortion objective during optimization), just to give them a fair chance. Lastly, the comparisons in Figure 3 used the *Base Hyperprior* model trained with $\lambda = 0.0025$.

# S4    Additional Results

On Kodak, we achieve the following BD rate savings: 17% for *SGA+BB* and 15% for *SGA* relative to *Base Hyperprior*; 17% for *SGA+BB* and 15% for *SGA* relative to BPG. On Tecnick, we achieve the following BD rate savings: 20% for *SGA+BB* and 19% for *SGA* relative to *Base Hyperprior*; 22% for *SGA+BB* and 21% for *SGA* relative to BPG.

Our experiments were conducted on a Titan RTX GPU with 24GB RAM; we observed about a $100\times$ slowdown from our proposed iterative inference methods (compared to standard encoder network prediction), similar to what has been reported in [Campos et al., 2019]. Note that our proposed standalone variant [M1] with SGA changes only compression and does not affect *decompression* speed (which is more relevant, e.g., for images on a website with many visitors).

Below we provide an additional qualitative comparison on the Kodak dataset, and report detailed rate-distortion performance on the Tecnick [Asuni and Giachetti, 2014] dataset.

(a) Original image
(kodim19 from Kodak)

(b) BPG 4:4:4;
0.143 BPP (PSNR = 29.3)

(c) Proposed (SGA);
0.142 BPP (PSNR = 29.8)

(d) [Minnen et al., 2018];
0.130 BPP (PSNR = 28.5)

Figure S4: Qualitative comparison of lossy compression performance on an image from the Kodak dataset. Figures in the bottom row focus on the same cropped region of images in the top row. Our method (c; [M1] in Table 1) significantly boosts the visual quality of the *Base Hyperprior* method (d; [M3] in Table 1) at similar bit rates. Unlike the learning-based methods (subfigures c,d), the classical codec BPG (subfigure b) introduces blocking and geometric artifacts near the rooftop, and ringing artifacts around the contour of the lighthouse.

Figure S5: Rate-distortion performance comparisons on the Tecnick [Asuni and Giachetti, 2014] dataset against existing baselines. Image quality measured in Peak Signal-to-Noise Ratio (PSNR) in RGB; higher values are better.

Figure S6: BD rate savings (%) relative to BPG as a function of PSNR, computed from R-D curves (Figure S5) on Tecnick. Higher values are better.

## Footnotes

[1]https://github.com/tensorflow/compression/blob/master/examples/bmshj2018.py

[2] `https://www.compression.cc/2018/challenge/`