[Reviews · NeurIPS 2020]

Review 1

Summary and Contributions: 1. Authors propose Stochastic Gumbel Annealing (SGA) method - a novel way of relaxing objectives involving discretized representations which results in improved optimization. Significance: High 2. Authors propose a scheme for using bits-back coding in the context of lossy compression with hierarchical autoencoders which doesn't yield large empirical improvements but is nevertheless an interesting methodological contribution. Significance: Medium 3. Authors perform a clean and comprehensive empirical investigation of the influence of different methods of optimizing the quantized representation at test/compression time, and a good ablation study of their proposed method. Significance: Medium/High. To my best knowledge and judgement, all 3 of them are novel and valuable. POST REBUTTAL: Thanks to the authors for quite thoroughly addressing my questions, I think it's a good submission, and I keep my score fixed at 7.

Strengths: Described in the contributions section.

Weaknesses: I have some concerns about the computational cost of the methods outlined above but it doesn't decrease the value of the work.

Correctness: To my best knowledge the claims and method is correct and the empirical methodology follows standard practice in the subfield.

Clarity: The paper is well organized and the exposition is clear. The paper provides a good structure of thinking about the problem of inference for compression by breaking down the sources of approximation gap and addressing each aspect individually. To me, this clarity becomes another contribution of the paper. Supplementary materials provide a good amount of implementation and experimental details, code was provided.

Relation to Prior Work: I think that the results are significant to the community interested in learning discrete representations, and of interest to wider NeurIPS community. To my best knowledge and judgement, all 3 of the contributions mentioned above are novel and valuable.

Reproducibility: Yes

Additional Feedback: Questions to authors: - This paper uses SGA only at test/compression time. Can SGA be used at training time as well? If so, would you expect that training the network with SGA would change the representation learned to yield even better results than those offered in this paper? Can you comment on that? - Could you provide some approximate timing comparison between different models? Particularly M1-3 A5 A6, although the more the better. I think the paper deserves some discussion on the trade-off between the amount of computation performed and the RD performance. I know it's such a cliche point to raise, but it's surely of interest to many. Your work is clearly valuable even if the computational considerations do not favour practical use of iterative inference or BB at test time, and any further information on this aspect won't affect my score negatively. Clarity: - fix the sentence l129-130 - mention an illustration of SGA relaxed objective in Section S1 in the main text, it's a good illustrative example


Review 2

Summary and Contributions: This paper improves the learned lossy image compression by enhancing the inference stage. Specifically, it introduces a new quantization method and an improved entropy coding model.

Strengths: - The ideas to improve the inference of learned image compression is very interesting. - The proposed methods are straightforward and sound. - The experimental results are extensive and comprehensive.

Weaknesses: - The proposed methods are based on previous iterative compression [Campos et al., 2019] and may lack of novelty. - Besides, the improvements from bitsback coding are relatively marginal, which means the main contribution comes from the quantization and novelty may be not sufficient for neurips.

Correctness: The claims are correct and the evaluations are comprehensive.

Clarity: The paper is well written and easy to follow.

Relation to Prior Work: Provide provides the differences between the proposed method and [Campos et al., 2019] in the implementation stage.

Reproducibility: Yes

Additional Feedback: - Is it possible to implemnt the proposed method based on the M4(context+hyperprior)? It would be better to provide the experimental results. - It is encourage to compare the proposed with the recent work in ECCV20[R1], where the whole encoder is updated for better RD performance. Is it possible to use this method to improve the inference for image compression? [R1] Guo Lu, Chunlei Cai, Xiaoyun Zhang, Li Chen, Wanli Ouyang, Dong Xu, and Zhiyong Gao. "Content Adaptive and Error Propagation Aware Deep Video Compression." arXiv preprint arXiv:2003.11282 (2020).


Review 3

Summary and Contributions: This paper proposed several improved techniques for lossy compression with a deep latent variable model, i.e. a hierarchical variational autoencoder. The authors conduct a comprehensive analysis on the amortization gap and the discretization gap which could be the key bottleneck on bridging the variational inference and lossy compression. And the proposed SGA and lossy bits back coding have achieved impressive empirical results on the compression benchmarks. POST REBUTTAL: Thanks to authors for addressing my concerns and provide new sensitive analysis results. I think the paper is above the acceptance threshold and I vote for acceptance.

Strengths: Exploring the potential of the generative model in compression is an important research direction. The proposed method is well motivated. The introduced SGA method is novel, which could optimize the rate-distortion objective with a small discretization gap. Though bits-back coding has been explored in the lossless compression scenario with latent variable models, it is nice to see the attempt in lossy compression settings. The experiment settings are extensive and the results are solid and impressive. And the overall writing is sound.

Weaknesses: The paper introduces two components in the paper. The stochastic Gumbel Annealing (SGA) tends to approximate the discretization with an annealing scheme. The main concern lies in how the model performance is influenced by the choice of annealing scheme. It would be better to add more details and sensitive analysis to show the effect of temperature annealing. The lossy bits back coding part may also lack some details: 1. How the entropy coding is implemented, i.e., which exact coding scheme is used? 2. The bits back coding saves the bitlength by getting the uniform bits during sampling of latent variable back at decoder part which could be used to transmit the so-called "side information". My question is how the side information is designed? If the side information refers to the encoded results of the previous image, then the benefit is related to the sequence length. Then how the sequence length is selected. If not, then what is the exact side information encoded and how the comparison between the proposed methods and other baseline is conducted. Besides, the benefit of lossy bits back coding seems to be marginal as shown in Figure 3. Another question is whether the proposed procedure is only compatible with the hierarchical vae? Maybe the lossy bits back coding is not easy to apply in the vanilla vae, but it will be a benefit to show the performance of SGA in a vanilla vae setting.

Correctness: yes

Clarity: yes

Relation to Prior Work: yes

Reproducibility: Yes

Additional Feedback: The SGA need the gumbel trick to backpropagate the gradient, it may be helpful to try some low variance estimator such as VIMCO[1]. [1]Variational inference for monte carlo objectives


Review 4

Summary and Contributions: In this paper, a new inference scheme is proposed to improve the performance of lossy image compression with deep latent variable models. The proposed scheme only changes inference at test time, therefore existing model structures and other innovations can be easily combined. The proposed method results in improved amortization, improved discretization and improved entropy coding.

Strengths: 1. Overall, the paper is well written. The proposed inference scheme is well described. 2. Both qualitative and quantitative comparisons are provided and showed that the proposed method improved compression quality. 3. A good ablation study is provided to justify the effectiveness of each part of the model.

Weaknesses: 1. In all the illustration and the equations, there is a clear definition of two hierarchies in the latent space: latent variable y and hyperlatents z. It would be more convincing if the author could provide some reasoning behind the choice of the latent structure. I am not sure why the latent structure is restricted to a 2-layer structure. Recent progress in VAE community including ladderVAE and structuredVAE shows that either a ladder structure with many hierarchies or a BN as the latent structure leads to better modelling performance. I would like the authors to explain if the proposed method can be applies to these innovations as well. 2. The model combines several existing techniques together, including iterative inference, Gumbel annealing, etc. Although the combination of these techniques seem to be working for the compression task, the overall novelty seems to be incremental.

Correctness: Yes.

Clarity: Yes.

Relation to Prior Work: Yes, the authors specifically discussed the difference between this method with related works. However, as mentioned before, this paper seems to be combining the innovations from several other papers.

Reproducibility: Yes

Additional Feedback: After the rebuttal, I believe the authors have addressed most of the concerns. So I will stay on the positive side to suggest accept the paper.

[Author Response · NeurIPS 2020]

We thank all reviewers for their thorough reviews. Given their already positive comments, we hope our responses below
will help increase the reviewers' confidence further and resolve any remaining doubts.

We'd like to remind reviewers that our proposals significantly improved the baseline and advanced the state-of-the-art on
the extremely competitive task of image compression. Building a successful data compression method on the progress
in generative modeling is nontrivial, and knowledge transfer between the two fields has only begun recently. Both our
annealed optimization method for integer representations and lossy bitsback are significant novel ideas in this direction.

**R1:** *"Can SGA be used at training time as well?"* → It can in principle, and should increase performance further (see
concurrent work [arXiv:2006.09952]) but a naive implementation would slow down training considerably. Mitigating
this by genrealizing ideas from [Kim et al., 2018] or [Marino et al., 2018] to SGA would be interesting followup work.

**R1:** *[timing comparison]* → Yes, that's a good point. We will provide detailed results in the final version of our paper.
Unfortunately, the rebuttal period was too short this year to generate a full analysis in time. In preliminary results, we
see a slowdown for *encoding* (i.e., compressing) of about 100x in our non-optimized code, which is similar to what has
been reported in [Campos et al., 2019]. Please note that our proposed standalone variant [M1] changes only compression
and does not affect *de*compression speed (which is more relevant, e.g., for images on a website with many visitors).

**R2 & R4:** *[limited novelty]* → We respectfully disagree. Our paper proposes two significant novel inventions: (i) a
novel inference method over an *infinite discrete* set, which significantly improves compression performance, and (ii) the
first lossy bitsback coding algorithm. We believe that each of these two would already be a significant contribution
on its own. However, we decided to combine both contributions into a single paper since, empirically, they strongly
complement each other (compare model [M1] to ablation [A3] in Section 4).

**R2 & R3:** *"improvements from bitsback coding are relatively marginal"* → We would like to clarify that these
improvements ([M2] in Section 4) are *on top of* an already novel method [M1] proposed in our paper, which already
improves performance significantly over the previous state of the art on this very competitive lossy image compression
benchmark. Further, we would like to point out that generalizing bitsback coding to lossy compression is nontrivial (this
has also recently been confirmed to us in private conversations with leading industry researchers working on this topic).
To the best of our knowledge, our work is the first empirically successful lossy variant of the 30-year-old bitsback
algorithm. We expect it to enable research on much more powerful hierarchical prior models for neural compression.

**R2 & R4:** *"implement the proposed method based on [M4] (context+hyperprior)"*, *"why the latent structure is
restricted to a 2-layer structure"* → Our paper focuses on *inference* rather than model architectures. While the proposed
inference methods can also be applied to other models, [M4] faces computational difficulties due to its inherently serial
nature (and is thus also excluded by [Johnston et. al, 2019]), and models from the broader VAE literature are often not
good for compression (e.g., compression models usually need much larger latent spaces). We deliberately used a model
that is common in the neural compression literature so that we could study the effect of improving inference in isolation
from improving the model architecture. We find such separation of concerns essential for generating scientific insights.

**R2:** *[comparison to arXiv:2003.11282]* → Thank you for the bringing this work to our attention, we will cite it in our
paper. The idea of optimizing the encoder parameters indeed seems related to our approach. By contrast, our approach
directly optimizes the output of the encoder, and is thus not limited by the expressivity of an encoder architecture

**R3:** *[analysis of temperature annealing]* → Good point! We will add the below curves to Figure 2 of the paper.
The left plot shows the true R-D objective of SGA us-
ing a naive temperature schedule $\tau(t) = 0.5e^{-ct}$, for
various decay factors $c$. As can be seen, too fast an-
nealing with this naive schedule can lead to suboptimal
solutions. The right plot shows that we can overcome
the suboptimality from fast annealing by fixing the tem-
perature to $\tau_0$ for some initial steps (until the R-D objective roughly converges) before annealing; we used $\tau_0 = 0.5$ as
it approximates soft quantization As shown, our resulting method is robust to different choices of the annealing factor $c$.

**R3:** *"How the entropy coding is implemented [...] how the side information is designed?"* → Like the original (lossless)
bitsback algorithm, the proposed lossy bitsback algorithm builds *on top of* entropy coding and is agnostic to both the
specific entropy coder used and the origin of the side information. We will provide a simple ANS entropy coder in our
public code repository. Our results for the proposed bitsback method [M2] report expected net bitrate for a *random
bitstring* of side information (this is a worst-case scenario since a random bitstring cannot be further compressed).

**R3:** *[SGA & bitsback in a non-hierarchical VAE]* → Thank you for pointing this out. Indeed, SGA does not strictly
require a hierarchical VAE, but hierarchical VAEs have proved to lead to superior compression performance in the
literature. The proposed lossy bitsback coding algorithm also builds on a hierarchical model; exploiting the increased
expressivity of a hierarchical model without paying the price of the marginalization gap is precisely its strength.

[Meta-Review · NeurIPS 2020]

*PROS: a clean and comprehensive empirical investigation of the influence of different methods of optimizing the quantized representation at test/compression time, and a good ablation study of their proposed method. A good ablation study is provided to justify the effectiveness of each part of the model *CONS: improvements from bitsback coding are relatively marginal Meta-reviewer recommendation: All reviwers voted for acceptance. I recommend the authors to look at the reviewers' comments to improve the paper for the camera ready version.